# Diagnostic Accuracy of Ischemia-Modified Albumin for Acute Coronary Syndrome: A Systematic Review and Meta-Analysis

**DOI:** 10.3390/medicina58050614

**Published:** 2022-04-28

**Authors:** Hyungoo Shin, Jae-Guk Kim, Bo-Hyoung Jang, Tae-Ho Lim, Wonhee Kim, Youngsuk Cho, Kyu-Sun Choi, Min-Kyun Na, Chiwon Ahn, Juncheol Lee

**Affiliations:** 1Department of Emergency Medicine, Hanyang University College of Medicine, Seoul 04763, Korea; seodtst@gmail.com (H.S.); erthim@gmail.com (T.-H.L.); jclee0221@gmail.com (J.L.); 2Department of Emergency Medicine, College of Medicine, Hallym University, Chuncheon 24253, Korea; gallion00@gmail.com (J.-G.K.); wonsee02@gmail.com (W.K.); faith2love@hanmail.net (Y.C.); 3Department of Preventive Medicine, College of Korean Medicine, Kyung Hee University, Seoul 02447, Korea; 4Department of Neurosurgery, Hanyang University College of Medicine, Seoul 04763, Korea; vertex-09@daum.net (K.-S.C.); mavmav@hanmail.net (M.-K.N.); 5Department of Emergency Medicine, College of Medicine, Chung-Ang University, Seoul 06974, Korea; cahn@cau.ac.kr

**Keywords:** meta-analysis, ischemia-modified albumin, acute coronary syndrome, emergency department, diagnostic test accuracy

## Abstract

The diagnostic usefulness of ischemia-modified albumin in acute coronary syndrome (ACS) has been questioned. The goal of this systematic review and meta-analysis was to see how accurate ischemia-modified albumin (IMA) was in diagnosing ACS in patients admitted to emergency departments (EDs). We searched for relevant literature in databases such as MEDLINE, EMBASE, and the Cochrane Library. Primary studies that reliably reported on patients with symptoms suggestive of ACS and evaluated IMA on admission to emergency departments were included. The QUADAS-2 tool was used to assess the risk of bias in the included research. A total of 4,761 patients from 19 studies were included in this systematic review. The sensitivity and specificity were 0.74 and 0.40, respectively, when the data were pooled. The area under the curve value for IMA for the diagnosis of ACS was 0.75, and the pooled diagnostic odds ratio value was 3.72. Furthermore, ACS patients with unstable angina had greater serum IMA levels than those with non-ischemic chest pain. In contrast to prior meta-analyses, our findings suggest that determining whether serum IMA levels are effective for diagnosing ACS in the emergency department is difficult. However, the accuracy of these findings cannot be ascertained due to high heterogeneity between studies.

## 1. Introduction

The most common cause of death worldwide is acute coronary syndrome (ACS) [1]. The diagnostic and clinical assessment of individuals admitted to emergency departments (EDs) with suspected ACS is difficult [2,3]. Biomarkers are critical in the diagnosis and treatment of ACS patients [4]. The elevation of cardiac troponin (cTn), a recognized marker for myocardial damage, is a component of the global definition of acute myocardial infarction (AMI) [1]. However, cTn concentrations and electrocardiograms may be normal in patients with suspected ACS [5]. Within the early hours of myocardial infarction, cTn does not achieve optimal sensitivity [6]; delaying sampling until 10–12 h following the onset of symptoms is recommended by guidelines [7]. Therefore, even in the absence of necrosis or prior to cTn rises, a sensitive biomarker of myocardial ischemia is required. Moreover, as global life expectancy increases, the prevalence and clinical consequences of frailty will rapidly grow. Given the scarcity of robust biomarkers of frailty, there is an increasing need for research targeted at developing novel and reliable diagnostic tools [8].

Ischemia-modified albumin (IMA) has been demonstrated to be a helpful biomarker for ACS in previous studies [9,10]. The albumin cobalt binding (ACB) test can identify IMA. Myocardial ischemia affects albumin’s N-terminus, lowering cobalt’s capacity to bind to albumin. The concentration of IMA rises within minutes after the beginning of ischemia, stays increased for 6–12 h, and then returns to normal after 24 h [11]. This quick IMA release could aid in covering the “troponin-blind” phase.

According to a recent meta-analysis, IMA is a highly sensitive biomarker for AMI and can rule out AMI when its value is normal in the ED setting [12]. However, the quality and interpretation of findings may be limited due to substantial heterogeneity (I^2^ = 87%). In contrast to the results of previous meta-analyses, several studies have shown that the use of IMA does not effectively exclude ACS [13,14]. Due to the mixed results, a systematic review and meta-analysis were needed to determine the diagnostic accuracy of IMA in patients admitted to the ED with suspected ACS.

## 2. Materials and Methods

### 2.1. Protocol and Registration 

The Preferred Reporting Items for Systematic Reviews and Meta-Analysis (PRISMA) guidelines were followed in this study, and the study was carried out in compliance with the Systematic Reviews of Diagnostic Test Accuracy principles [15,16]. The protocol was registered at http://www.crd.york.ac.uk/PROSPERO/ (CRD42019139197), accessed on 16 December 2019. This study did not infringe patients’ personal information or rights and therefore there was no need for ethical approval.

### 2.2. Eligibility Criteria

#### 2.2.1. Study Design

Relevant studies that (1) reported data from patients with symptoms suggestive of ACS, (2) tested serum IMA levels upon patient admission to the ED, or (3) assessed the diagnostic performance of IMA for ACS were included in our meta-analysis.

#### 2.2.2. Participants

Adult patients with suspected ACS who presented to the ED using IMA were included in our study population. Studies with healthy subjects as a control group and no clinical signs of cardiac disease were excluded.

#### 2.2.3. Index Tests

Only studies assessing the diagnostic accuracy of IMA assessed in blood samples collected upon a patient’s admission to the ED were included in this study. The IMA index tests included the ACB test only. Studies in which serum IMA levels were reported in absorbance units or enzyme-linked immunosorbent assays were excluded.

#### 2.2.4. Reference Tests

ACS refers to a group of disorders that range from unstable angina (UA), which is associated with reversible myocardial cell injury, to AMI, which is associated with irreversible myocardial necrosis [17]. AMI was defined in accordance with a standard definition. Most studies incorporated clinically diagnosed ACS and electrocardiogram abnormalities as composite diagnostic reference standards or a diagnostic standard for ACS. We attempted to collect the data from the included studies for ACS definition, as indicated in Appendix A.

### 2.3. Information Sources and Literature Search Strategy

On 21 November 2020, two experienced reviewers (H.S. and J.-G.K.) conducted a literature search. The Ovid interface was used to search the MEDLINE (1946 to 17 November 2020) and EMBASE (1974 to 17 November 2020) databases, as well as the Cochrane Library (all years) (Appendix A). We also manually reviewed the references cited by all qualified studies to find more studies that were relevant. ‘Ischemia modified albumin’ and ‘myocardial infarction’ or ‘acute coronary syndrome’ or ‘coronary artery disease’ or ‘angina’ were used as search terms. Articles with prospective or retrospective observational studies were included.

### 2.4. Study Selection

All of the studies found in the literature search were organized using the reference management software Endnote X9 (Clarivate Analytics, Philadelphia, PA, USA). Two reviewers (H.S. and J.-G.K.) examined the title, abstract, and research type of each publication, and irrelevant studies were eliminated. Reviews, case reports, editorials, letters, comments, conference papers, meta-analyses, animal studies, duplicate studies, irrelevant populations (non-ACS patients), irrelevant control groups (healthy individuals as control group), and irrelevant outcome measures (main outcome or analysis not eligible for this meta-analysis) were all excluded. Studies that did not match the criteria for enrollment in our study or had inadequate data despite the authors having been contacted were also excluded. The reasons why potentially relevant studies failed to meet the eligibility criteria are presented in Appendix A. When the two reviewers disagreed, a third reviewer (B.-H.J.) stepped in and the discrepancies were discussed until a consensus was established. Using the same exclusion criteria, the full texts of the selected articles were retrieved, rescreened, and reviewed more thoroughly for eligibility. We retrieved the complete texts of the selected articles after eliminating ineligible research, which we rescreened and reviewed in detail using the same criteria.

### 2.5. Data Collection Process and Data Items

According to the Cochrane guidelines, two reviewers (H.S. and J.-G.K.) extracted data from the selected articles [18]. A third reviewer (B.-H.J.) further reviewed any undetermined disagreements between the two reviewers. The two reviewers obtained the characteristics and results of the selected studies. The following variables were extracted from all studies: author, year of publication, country, inclusion period, study design, and study population. Variables such as cut-off points (U/mL) for IMA were considered. Individual study findings were also obtained, including true-positive, false-positive, false-negative, and true-negative results. If a variable of interest was not described in the research, we emailed the corresponding author of each study for more information. Despite contacting the authors, studies that lacked data were eliminated from the meta-analysis.

### 2.6. Risk of Bias in Individual Studies

Two reviewers (H. Shin and J. Kim) independently assessed the methodological quality of the identified articles, blinded to authorship and journal. A checklist derived from the Quality Assessment of Diagnostic Accuracy Studies (QUADAS)-2 tool was used to examine patient selection, index test, reference standard, flow, and timing [19].

### 2.7. Statistical Analysis

The retrieved data were used to construct sensitivity and specificity point estimates, as well as 95% confidence intervals (CIs) for IMA for each primary study. The sensitivity, specificity, positive predictive value, and negative predictive value were calculated using true-positive, true-negative, false-positive, and false-negative rates (NPV). The summary estimates of the diagnostic performance measures (sensitivity, specificity, and positive and negative likelihood ratios) were calculated using bivariate mixed-effect regression model parameter estimates. The researchers calculated pooled sensitivities, specificities, diagnostic odds ratios (DORs), areas under the curve (AUCs), and positive and negative likelihood ratios. It was thought that features that showed a pooled DOR with a 95% CI that did not include 1 were informative. The entire prognostic performance of the IMA was summarized using summary receiver operating characteristic (SROC) curves, which also represented the determined value of Q* index and AUC [20,21]. I^2^ statistics were used to evaluate the amount of heterogeneity between articles attributable to genuine differences between studies (rather than differences due to random error or chance), with values of 25%, 50%, and 75% being considered low, moderate, and high, respectively [22].

In addition, the association between IMA and non-ischemic chest pain (NICP) and acute coronary syndrome (ACS) was examined. The standardized mean differences were used to assess the strength of the association between serum IMA levels and NICP and ACS (SMD). Considering the diversity of countries, medical systems, and inclusion periods, a random-effect model was employed to synthesize the individual data of the included studies [23]. The differences in serum IMA levels between the comparison groups were extracted as mean differences with a 95% CI.

Standard descriptive statistics were used to summarize the study features and retrieved covariates. Dichotomous variables were reported as frequencies (%), whereas continuous variables were reported as means (standard deviation (SD)). The statistical significance for hypothesis testing was set at 0.05 for two-tailed heterogeneity testing and 0.10 for two-tailed tests. The statistical analysis was performed using Meta-Disc software 1.4 (Clinical Biostatistics, Ramony Cajal Hospital, Madrid, Spain) and RevMan version 5.3 (Cochrane Collaboration, Oxford, UK), with a *p*-value of <0.05 considered statistically significant.

### 2.8. Risk of Bias across Studies

The R package ‘meta’ was used to identify publication bias (R version 3.3.2). A funnel plot and Egger’s test were also used to evaluate it. The presence of bias was revealed by the asymmetry of the funnel plot and a *p*-value of <0.05 using Egger’s test.

### 2.9. Additional Analyses

Subgroups were analyzed by cut-off values (U/mL) of serum IMA levels (<85 vs. 85 vs. >85) according to the median value across the included studies, location of the country (Asia vs. Europe vs. North America), and target condition (ACS vs. AMI, UA vs. non-ST segment elevation myocardial infarction (NSTEMI) vs. ST segment elevation myocardial infarction (STEMI)). To assess the potential reasons for heterogeneity, sensitivity analyses and meta-regression analyses employing multiple covariates were performed.

Meta-regression analyses identify whether there is a significant association between an independent variable, such as the number of patients, and the prevalence of ACS. The *p*-value and regression coefficient (r) can be used to measure the strength of this association after constructing a regression model. If there is a substantial association, it is possible that the study variable is the cause of the observed variability [24,25].

## 3. Results

### 3.1. Study Selection

During the database search, a total of 681 records were identified (Figure 1) and 503 records were evaluated for eligibility after 178 duplicates were removed. Following this, 417 studies were removed after titles and abstracts were screened. There were 86 records identified as potentially relevant, and full-text articles were retrieved for a more thorough review. We excluded 67 studies for irrelevant outcome measures (*n* = 25), irrelevant populations (*n* = 24), irrelevant control groups (*n* = 10), conference abstracts (*n* = 6), data duplicated from the same study (*n* = 1), and reviews (*n* = 1). Finally, our meta-analysis included 19 studies, which enrolled 4761 patients [13,14,26,27,28,29,30,31,32,33,34,35,36,37,38,39,40,41,42].

### 3.2. Study Characteristics

The 19 studies included a total of 4761 patients, and the prevalence of ACS was 37.2% (1770 patients, range 6.9–75.9%) (Table 1). Serum IMA levels were determined by the ACB test (range of diagnostic threshold = 31.95–117 U/mL). The summary estimates of the diagnostic performance measures were derived from 17 studies [13,14,27,28,29,30,31,33,34,35,36,37,38,39,40,41,42]. AMI was the target condition in 3 studies, while ACS was the target condition in 14 studies. The strength of association of serum IMA levels with ACS and NICP was measured by the standardized mean differences with 11 studies [13,14,26,28,29,32,33,35,38,39,41].

### 3.3. Risk of Bias within Studies

All included studies were evaluated to determine the risk of bias and applicability concerns (Appendix A). The risk of bias for patient selection and index tests were assessed as high risk in three and eight studies, respectively. In addition, five studies had a high risk for flow and timing bias. In the patient selection, index test, and reference standard domains, all studies demonstrated a poor level of applicability to our research question.

### 3.4. Results of Meta-Analyses

#### 3.4.1. Diagnostic Performance of Serum IMA Level for the Diagnosis of ACS

The 17 studies that focused on ACS or AMI as a target condition were included in summary estimates for the diagnostic performance of ACS (Table 2). The serum IMA level ranged from 0.40 to 0.93 for sensitivity and from 0.08 to 0.88 for specificity (Appendix A). The pooled sensitivity and specificity were 0.74 (95% CI = 0.71–0.76; I^2^ = 90.4%) and 0.40 (95% CI = 0.38–0.42; I^2^ = 97.2%), respectively. The pooled positive and negative likelihood ratios were 1.63 (95% CI = 1.33–2.00; I^2^ = 93.2) and 0.49 (95% CI = 0.35–0.69; I^2^ = 90.3), respectively. The pooled DOR value of the serum IMA levels for the diagnosis of ACS was 3.72 (95% CI = 2.00–6.91; I^2^ = 91.0%). The AUC value was 0.75 (SE = 0.05, Q = 0.69) in 17 studies (Figure 2). Subgroup analysis was performed using the cut-off value of serum IMA levels (<85 vs. 85 vs. >85). As the cut-off value of serum IMA levels increased, the pooled sensitivity and pooled AUC decreased (Table 2).

#### 3.4.2. Comparing Serum IMA Levels between Patients with ACS and NICP

Serum IMA levels were relatively higher in patients with ACS than in those with NICP, demonstrating a positive association (11 studies; SMD = 1.30; 95% CI = 0.59–2.00; I^2^ = 98%; *p* < 0.001; Appendix A). Additionally, serum IMA levels were relatively higher in patients with ACS than in those with NICP in six studies from Asia (SMD = 2.01; 95% CI = 0.68–3.34; I^2^ = 99%; *p* = 0.003).

Serum IMA levels were relatively higher in patients with UA than in those with NICP (five studies for UA; SMD = 1.20; 95% CI = 0.33–2.08; I^2^ = 97%; *p* = 0.007, Appendix A). There were no statistically significant differences in serum IMA levels between patients with NSTEMI (*p* = 0.23) and STEMI (*p* = 0.06) and those with NICP.

### 3.5. Additional Analyses

Sensitivity analyses and meta-regression analyses were performed to investigate the potential causes of heterogeneity; the additional analyses showed high heterogeneity (Appendix A). No significant association was observed between serum IMA levels and sources of heterogeneity, such as the number of patients or the prevalence of ACS.

### 3.6. Publication Bias

There was no definite asymmetry in the forest plot. We did not observe any publication bias in studies concerning the relationship between serum IMA levels in patients with ACS and in those with NICP, based on Egger’s regression test (*p* = 0.217; Figure 3).

## 4. Discussion

The sensitivity and specificity of serum IMA were estimated to be 0.74 and 0.40, respectively, in this meta-analysis. The pooled DOR value for the serum IMA level was 3.72, and the AUC value was 0.75 for the ACS diagnosis. Although serum IMA levels were relatively higher in patients with ACS than in those with NICP, the sensitivity and specificity of serum IMA were estimated to be 0.74 and 0.40, respectively, in this meta-analysis. Insufficiently high sensitivity and the low specificity of serum IMA level for ACS diagnosis limit the practical value of the test.

IMA, as determined by the ACB test, is a biomarker approved by the US Food and Drug Administration for the purpose of excluding ACS [10,43]. In the ischemia condition, free radicals transiently modify the N-terminal sequence of human albumin, reducing albumin’s ability to bind cobalt, hence serum IMA levels can be high [44,45]. Following the onset of myocardial infarction, serum IMA levels rise rapidly and return to baseline after 24 h [46]; during the cTn delayed release interval, known as the ‘troponin-blind’ period, this rapid-release of IMA allows for an early diagnostic opportunity. IMA demonstrated the highest NPV (0.96) in a previous trial involving 256 patients who presented to the ED within three hours of the beginning of clinical signs and symptoms consistent with ACS [30]. In another study, IMA had a greater NPV (0.92) than the combination of myoglobin, CK-MB, and cTnT in 413 patients with suspected ACS [37]. IMA seems to add relevant diagnostic value, especially when combined with cTn and electrocardiogram. A meta-analysis comprising eight studies involving 1812 patients showed that IMA had an NPV of 0.91 [10]. These findings suggest that IMA may be a useful early diagnostic marker for excluding ACS in patients with chest pain. IMA was found to be substantially associated with left ventricular ejection fraction and served as an early indicator of left ventricular dysfunction in STEMI patients [47]. Ischemia reperfusion injury to the myocardium promotes the generation of reactive oxygen species, resulting in oxidative stress [48]. IMA is mostly formed as a result of the oxidative stress response induced by ischemic reperfusion injury [49,50]. The results from Chen et al. support this mechanism and showed that elevated oxidative stress can result in increasing serum IMA levels, and increased oxidative stress increases the likelihood of coronary collateral circulation forming [51].

The release of cardiac markers is time-dependent and an initial negative result does not exclude the presence of myocardial ischemia [32]. The diagnostic value of serum IMA levels could be affected by the time interval between the onset of chest pain and ED arrival. Although most studies in this meta-analysis included patients whose chest pain episode occurred a few hours prior to their arrival at EDs, the time interval is inconsistent; this meta-analysis did not identify the association between the diagnostic accuracy of serum IMA levels for ACS diagnosis and the heterogeneity of the study population. Apple et al. showed that serum IMA levels did not increase in the period immediately after heavy physical exercise but after 24–48 h [52]. This latent increase may be contributed by gastrointestinal ischemia or delayed response to skeletal muscle ischemia. This may potentially complicate use of the test in clinical practice.

Recent studies indicate that IMA is specific neither for myocardial ischemia nor for infarction [35,53]. IMA is mostly formed as a result of the oxidative stress response induced by ischemic reperfusion injury, which includes not only cardiac but also extracardiac events. Problems with the lack of cardio-specificity of IMA have been also reported [46]; elevated serum IMA levels may occur in acute stroke [54], pulmonary embolus [55], and end-stage renal disease [56]. Serum IMA concentrations, representing protein oxidative damage, were considerably greater in morbidly obese patients than in healthy women [57]. Moreover, circulating IMA showed variation according to circadian rhythm, with a negative correlation with melatonin levels in patients with STEMI [58]. Severe hypoalbuminemia may also affect the result of an ACB test, which will cause false-high result [59].

According to this meta-analysis, as the cut-off value of serum IMA level increased, the pooled AUC decreased. Emergency physicians caring for older patients encounter diagnostic difficulties due to the increased sensitivity but poor specificity of IMA testing. The optimal serum IMA cut-off value for ruling out ACS may vary significantly with age; older patients may have higher cut-off levels. The optimal cut-off value of IMA may also be influenced by comorbidities. Different serum IMA cut-off values can be applied according to patients’ comorbidities, such as end-stage renal disease [49]. IMA values should be interpreted with caution, in consideration of patients’ symptoms and comorbidities. The lack of cardio-specificity remains a major deterrent against its application. IMA may be a useful supplementary tool for assessing patients with cardiac chest pain for the emergency physician but a better understanding of this biomarker is necessary [60].

There are several limitations to this meta-analysis. First, inconsistencies in the study’s statistical and clinical data were not resolved. Although we conducted sensitivity and meta-regression analyses to ascertain the possible sources of heterogeneity, the additional analyses showed high heterogeneity. This may restrict the quality and interpretation of the findings. This heterogeneity may have been caused by differences between studies in the inclusion criteria, diverse cut-off values of serum IMA levels, the location of the country, and target conditions, such as ACS vs. AMI. The high heterogeneity may be influenced by the severity of patients and the degree of myocardial ischemia. Decreased left ventricular systolic function and a larger left ventricle could elevate serum IMA levels [47]. Patients’ characteristics, including age, sex, and comorbidities, which we have not yet identified due to insufficient data, may also have contributed to high heterogeneity. To resolve this heterogeneity, serum IMA should be examined for its diagnostic usefulness in ACS in well-designed studies, such as randomized clinical trials. While serum IMA levels may rise after myocardial ischemia, such as during vigorous exercise [52], IMA may be influenced by myocardial antioxidant capacity or increased shock protein levels [61]. Second, the majority of the studies included in this meta-analysis were observational and selection bias may have occurred. This may have led to the failure to apply appropriate eligibility criteria and adequately control for confounding factors. Third, this meta-analysis assessed studies with a high probability of bias. The proportion of data from studies with a high risk of bias is sufficient to influence the interpretation of studies. Fourth, this study demonstrated relatively low sensitivity, and various cut-off values were tested for obtaining a high diagnostic accuracy for IMA. To compensate for the effect of the cut-off value in each trial, we carefully employed diagnostic odds ratios that combined sensitivity and specificity since they are reasonably consistent regardless of variation in the threshold of the pooled analysis [62].

## 5. Conclusions

In contrast to previous analyses, there are several limitations to using serum IMA level as the sole marker for diagnosing ACS in the ED setting owing to the high degree of heterogeneity among studies included in this meta-analysis.

## Figures and Tables

**Figure 1 medicina-58-00614-f001:**
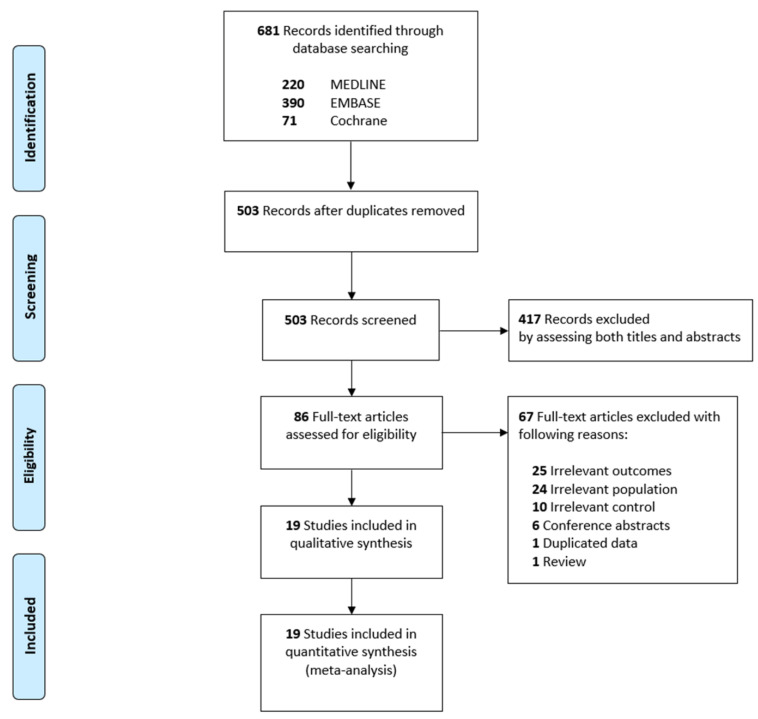
Flow diagram for the identification of relevant studies.

**Figure 2 medicina-58-00614-f002:**
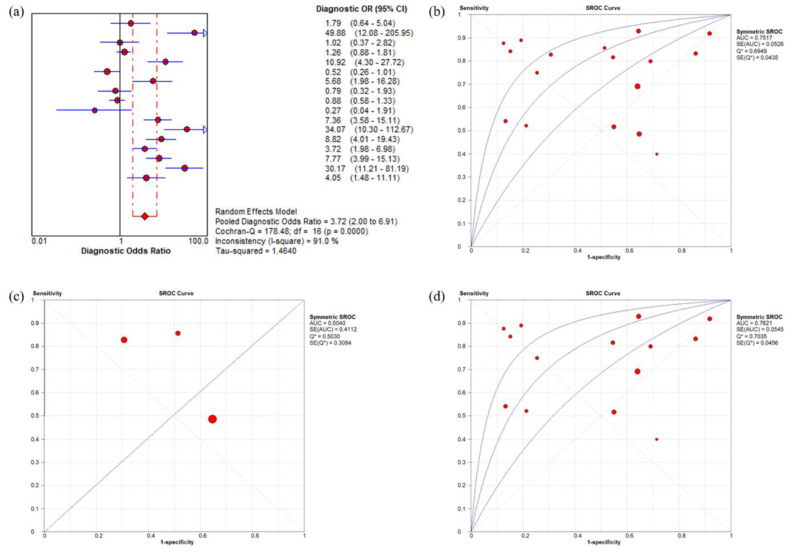
Forest plot of the DOR and the SROC curves of serum IMA levels for diagnosis of ACS. (**a**) The pooled DOR value of the serum IMA levels for diagnosis of ACS was 3.72. (**b**) The area under the SROC was 0.75 in 17 studies. (**c**) When limited to studies assessing AMI, the area under the SROC was 0.50 in three studies. (**d**) For studies that assessed ACS, including UA, the area under the SROC was 0.76 in 14 studies. DOR, diagnostic odds ratio; SROC, summary receiver operating characteristic; IMA, ischemia-modified albumin; ACS, acute coronary syndrome; UA, unstable angina.

**Figure 3 medicina-58-00614-f003:**
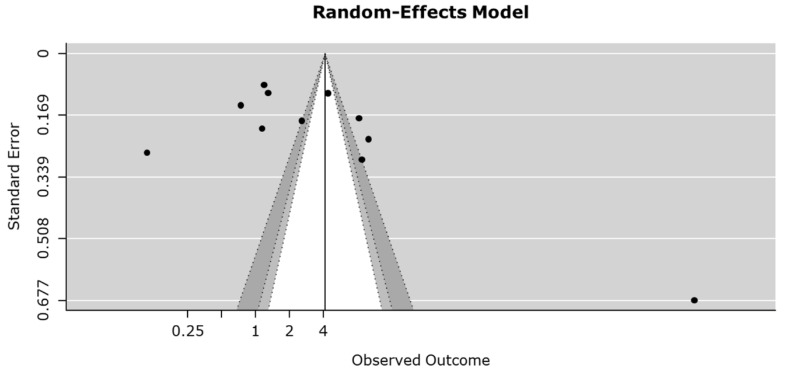
Funnel plot and Egger’s regression test to assess for publication bias.

**Table 1 medicina-58-00614-t001:** Study characteristics.

Authors	Year	Country	Inclusion Period	Study Design	Sample Size, n (%)					IMA Cut-Off (U/mL)
					Total	NICP	ACS			
							UA	NSTEMI	STEMI	
Aggarwal	2012	India	-	-	100	50 (50.0)	-	50 (50.0)		-
Anwaruddin	2005	USA	-	sPOS	193	168 (87.0)	16 (8.3)	9 (4.7)		90
Bhakthavatsala	2014	India	-	-	89	24 (27.0)	25 (28.1)	14 (15.7)	26 (29.2)	80
Bhardwaj	2011	USA	-	mPOS	318	256 (80.5)	40 (12.6)	22 (6.9)	-	85
Chapentier	2010	France	May 2006–Mar 2007	sPOS	677	492 (72.7)	77 (11.4)	99 (14.6)	-	85
Christenson	2001	USA	Jan 2000–Jun 2000	mROS	224	189 (84.3)	-	35 (15.7)		75
Collinson	2006	UK	-	sPOS	538	501 (93.1)	-	37 (6.9)	-	85
Gurumurthy	2014	India	-	-	540	135 (25.0)	135 (25.0)	135 (25.0)	135 (25.0)	84.4
Hjortshoj	2010	Denmark	Feb 2007–May 2007	-	107	72 (67.3)	-	35 (32.7)	-	88.2
Keating	2006	UK	Oct 2003–Feb 2005	mPOS	277	235 (84.8)	42 (15.2)		-	86
Kim	2010	Korea	Nov 2005–Aug 2007	sPOS	367	162 (44.1)	97 (26.4)	84 (22.9)		98.5
Kountana	2013	Greece	Mar 2010–Dec 2011	-	33	28 (84.8)	5 (15.2)	-		31.95
Lee	2007	Korea	Jun 2005–May 2006	-	413	284 (68.8)	129 (31.2)			85
Liyan	2009	China	Nov 2005–Oct 2006	sPOS	108	26 (24.1)	39 (36.1)	43 (39.8)		70.5
Roy	2004	UK	Dec 2000–Jun 2001	sPOS	131	67 (51.1)	48 (36.6)	16 (12.2)	-	93.5
Sinha	2004	UK	Jun 2001–Dec 2001	sPOS	208	77 (37.0)	85 (40.9)	26 (12.5)	20 (9.6)	85
Sokhanvar	2012	Iran	Jul 2009–Mar 2010	sPOS	226	106 (46.9)	98 (43.4)	22 (9.7)	-	85
Takhshid	2010	Iran	Aug 2008–Sep 2009	sPOS	123	53 (43.1)	45 (36.6)	25 (20.3)		82.4
Talwalkar	2008	USA	Dec 2004–Jan 2005	sROS	89	66 (74.2)	23 (25.8)			117

Abbreviations: NICP, non-ischemic chest pain; ACS, acute coronary syndrome; UA, unstable angina; NSTEMI, non-ST segment elevation myocardial infarction; STEMI, ST segment elevation myocardial infarction; IMA, ischemia-modified albumin; sPOS, single-center prospective observational study; mPOS, multicenter prospective observational study; mROS, multicenter retrospective observational study; sROS, single-center retrospective observational study.

**Table 2 medicina-58-00614-t002:** Pooled estimates for diagnostic accuracy of serum IMA level for acute coronary syndrome.

Target Condition	Studies, n	Pooled Sensitivity (95% CI)	Pooled Specificity (95% CI)	Pooled PLR (95% CI)	Pooled NLR (95% CI)	Pooled DOR (95% CI)	Pooled AUC (SE)
Total studies	17	0.74 (0.71–0.76) [I^2^ = 90.4%]	0.40 (0.38–0.42) [I^2^ = 97.2%]	1.63 (1.33–2.00) [I^2^ = 93.2%]	0.49 (0.35–0.69) [I^2^ = 90.3%]	3.72 (2.00–6.91) [I^2^ = 91.0%]	0.75 (0.05)
AMI	3	0.72 (0.63–0.80) [I^2^ = 86.6%]	0.45 (0.41–0.49) [I^2^ = 96.9%]	1.51 (0.74–3.11) [I^2^ = 94.8%]	0.49 (0.11–2.22) [I^2^ = 94.4%]	3.10 (0.41–23.69) [I^2^ = 93.8%]	0.50 (0.41)
ACS	14	0.74 (0.71–0.76) [I^2^ = 91.5%]	0.38 (0.36–0.40) [I^2^ = 97.3%]	1.65 (1.32–2.06) [I^2^ = 93.3%]	0.48 (0.34–0.69) [I^2^ = 89.7%]	3.89 (1.98–7.64) [I^2^ = 91.1%]	0.76 (0.05)
Cut-off (U/mL)							
<85	4	0.86 (0.81–0.90) [I^2^ = 55.0%]	0.73 (0.64–0.80) [I^2^ = 90.6%]	3.33 (1.26–8.79) [I^2^ = 79.8%]	0.28 (0.10–0.78) [I^2^ = 89.0%]	12.55 (2.04–77.21) [I^2^ = 86.6%]	0.91 (0.11)
=85	5	0.76 (0.73–0.79) [I^2^ = 94.4%]	0.35 (0.32–0.38) [I^2^ = 98.2%]	1.43 (1.10–1.86) [I^2^ = 94.7%]	0.51 (0.33–0.79) [I^2^ = 84.3%]	3.11 (1.30–7.39) [I^2^ = 89.5%]	0.73 (0.07)
>85	5	0.62 (0.57–0.67) [I^2^ = 87.2%]	0.47 (0.43–0.51) [I^2^ = 98.1%]	1.72 (1.04–2.84) [I^2^ = 94.7%]	0.59 (0.32–1.08) [I^2^ = 88.8%]	2.99 (0.93–9.62) [I^2^ = 91.6%]	0.67 (0.08)

Abbreviations: PLR, positive likelihood ratio; NLR, negative likelihood ratio; DOR, diagnostic odds ratio; AUC, area under the curve; SE, standard error; AMI, acute myocardial infarction; ACS, acute coronary syndrome.

## Data Availability

The datasets generated during the current study are available from the corresponding author upon reasonable request.

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
