# Peer review of "Diagnostic Accuracy of Ischemia-Modified Albumin for Acute Coronary Syndrome: A Systematic Review and Meta-Analysis"

_medicina, 2022, doi:10.3390/medicina58050614_

Round 1

Reviewer 1 Report

Report on the manuscript "Diagnostic accuracy of ischemia-modified albumin for acute 2 coronary syndrome: A systematic review and meta-analysis" by Hyungoo Shin et al.  The paper reports a statistical study to determine whether serum IMA level helps diagnose ACS in the ED. Although, the main disadvantage is that the accuracy of these findings cannot be assured because of the high heterogeneity among studies.

There are a few corrections that should be considered :

  1. The manuscript states in the introduction section, "several studies have shown that the use of IMA does not effectively exclude ACS..." however, the meta-analysis performed by the authors did not apport any different outcomes or new findings. So then, I advise carrying out a multivariate analysis with PCA to include other variables such as obesity, diabetes, age, sex, etc. This analysis will reduce the heterogeneity and offer, in this way, a better biomarker for ACS using IMA.

Study section, the authors need to describe the full criteria of exclusion—for example, irrelevant populations, irrelevant control groups, and irrelevant outcome measures.

  1. line 135. Include the numeric values for the cutt-off points for IMA.
  2. Line 188: units? Describe the values and units associated to 85 cut-offs.
  3. Figure 1. What criteria were used to select irrelevant populations for this analysis? Please explain this question in the study section

Author Response

We would like to express our sincere thanks to you and the reviewers for the thorough review of our manuscript (Manuscript ID medicina-1657873) titled “Diagnostic accuracy of ischemia-modified albumin for acute coronary syndrome: A systematic review and meta-analysis.” and for the opportunity to submit a revised and improved version. We believe that by addressing the concerns, we have considerably improved our manuscript. Below this letter, we have provided point-by-point responses to the reviewer’s comment.

Reviewer 2 Report

The article “Diagnostic accuracy of ischemia-modified albumin for the acute coronary syndrome: A systematic review and meta-analysis.” Focuses on the diagnostic ability of ischemia-modified albumin (IMA) for the ruling-out acute coronary syndrome (ACS) in ED. They concluded that serum IMA levels did not show high sensitivity and specificity which made it difficult to use for screening or diagnosing ACS. The work is quite well. But we still find some problems with the paper.

Major problems

  1. The idea of this article is less creative. Compared to previous works as mentioned in the article, Shin and colleagues provided no additional information about the IMA for ruling in or ruling out ACS in ED compared troponin test and showed the same result: the IMA did not useful for patients who had suspected ACS. The authors also want to show that the IMA might be an additional role for patients who had a ‘troponin-blind’ period. Unfortunately, IMA is quite the same as troponin to us. One of the most critical reasons for this result could be affected by the heterogeneity of the study population. Because of time-dependent accumulation, the time gap between the onset time of chest pain and ED arrival time could be important.

  1. Discussion about the usefulness or limitation of the IMA was superficial. The authors need to give a more in-depth possible explanation of pathophysiology when they try to insist on the usability of the biomarkers for screening certain diseases. For example, serum IMA might be increased by nonspecific conditions, such as severe hypoalbuminemia (Acta Med Indones. 2006;38(2):92-6). Moreover, the 5th paragraph (line 323 – 329) about the association between IMA and acute aortic dissection was a quite irrelevant issue about the topic. Therefore, the authors need to revise the whole paragraph in the discussion section for strengthening their findings.

Minor problems

  1. Some abbreviations, such as NICP (non-ischemic chest pain), ESRD (end-stage renal disease), and ECG (electrocardiogram) were not necessary because of unfamiliar or limited numbers of mentions (just 2 uses in the entire manuscript).

  1. Correct typos
  • Line 160: ‘gray-white matter ratio’ -> IMA

Author Response

(The authors gave the same response as above.)

Round 2

Reviewer 1 Report

The authors have reviewed all my comments and have answered or clarified my queries satisfactorily. As a result, the corrected manuscript has been substantially improved, but I advise extensively rewriting the conclusions section before publication.

Reviewer 2 Report

Thanks for the authors’ effort to improve quality. However, there were still some issues to enhance the meaning of the authors’ findings.

Correct typo:

Line 435: The sensitivity and specifity of serum IMA -> “the” sensitivity and ~

Line 442: In the ischemia condition, free radicals transiently modify the N-terminal sequence of hyman albumin, reducing albumin’s ability to bind cobalt., hence the high serum IMA levels [43,44] -> delete “.” & hence the serum IMA levels can be high.

Line 446. IMA was found to be substantially associated with left ventricular ejection fraction and served as an early indicator of left ventricular dysfunction in STEMI patients. [50]. -> delete “.”

Line 520-523. As global life expectancy increases, the prevalence and clinical consequences of frailty will rapidly grow. Given the scarcity of robust biomarkers of frailty, there is an increasing need for research targeted at developing novel and reliable diagnostic tools. -> This content is quite general which is not suitable for discussion section. It would be better to mentioned in introduction section.

Line 526-529: IMA values should be interpreted cautiously, with consideration clinical circumstances and comorbidity. The lack of cardio-specificity remains the major problem for clinical use. IMA may be to the emergency physician seessing patients with cardiac chest pain, but we require a better understanding on this marker. -> These sentences need to revise with english editing service. The rationale was not solid and not formal.